# Preoperative Diagnosis and Indications for Endoscopic Resection of Superficial Esophageal Squamous Cell Carcinoma

**DOI:** 10.3390/jcm10010013

**Published:** 2020-12-23

**Authors:** Katsunori Matsueda, Ryu Ishihara

**Affiliations:** Department of Gastrointestinal Oncology, Osaka International Cancer Institute, 3-1-69 Otemae, Chuo-ku, Osaka 541-8567, Japan; rge43gs@yahoo.co.jp

**Keywords:** superficial esophageal squamous cell carcinoma, endoscopic resection, endoscopic submucosal dissection, endoscopic treatment

## Abstract

Endoscopic resection (ER) is the mainstay of treatment for superficial esophageal squamous cell carcinoma (SESCC) instead of esophagectomy because of its minimal invasiveness and favorable clinical outcomes. Developments in endoscopic submucosal dissection have enabled en bloc resection of SESCCs regardless of size, thus reducing the risk of local recurrence. Although ER for SESCC is effective, metastasis may subsequently occur. Additionally, extensive esophageal ER confers a risk of postoperative esophageal stricture. Therefore, accurate assessment of the invasion depth and circumferential extent of SESCCs is important in determining the indications for ER. Diagnostic accuracies for SESCC invasion differ between epithelial (EP)/lamina propria (LPM), muscularis mucosa (MM)/submucosal (SM1), and SM2 cancers. ER is strongly indicated for clinically diagnosed (c)EP/LPM cancers because 90% of these are as pathologically diagnosed (p)EP/LPM, which has a very low risk of metastasis. Remarkably, the diagnostic accuracy for cMM/SM1 differs significantly with lateral spread of cancer. Eighty percent of cMM/SM1 cancers with ≤3/4 circumferential spread prove to be pEP/LPM or pMM/SM1, which have very low or low risk of metastasis. Thus, these are adequate candidates for ER. However, given the relatively low proportion of pEP/LPM or pMM/SM1 and high risk of subsequent stricture, ER is not recommended for whole circumferential cMM/SM1 cancers. For cMM/SM1 cancers that involve >3/4 but not the whole circumference, ER should be considered on a lesion-by-lesion basis because the risk of post-ER stricture is not very high, but the proportion of pEP/LPM or pMM/SM1 is relatively low. ER is contraindicated for cSM2 cancers because 75% of them are pSM2, which has high risk of metastasis.

## 1. Introduction

Esophageal cancer is the sixth most common cause of cancer-related mortality, with 455,800 new cases and 400,200 deaths worldwide in 2012 [1]. Although the survival rate of patients with esophageal cancer remains poor, they can potentially be cured by esophagectomy, endoscopic resection (ER), or chemoradiotherapy (CRT) if diagnosed at an early stage [2,3,4,5,6,7,8]. Esophagectomy is historically the standard treatment for patients with superficial esophageal squamous cell carcinoma (SESCC). However, this procedure is only possible in patients who can tolerate the procedure, and it is associated with significant mortality and substantial morbidity [9,10]. ER is a safer and less invasive procedure and better preserves esophageal function than surgical resection or CRT. ER has a high local cure rate for Stage I esophageal cancer [5] but not for metastases. Endoscopic submucosal dissection (ESD), a variation of ER, was widely implemented approximately 20 years ago [11]. Recent improvements in performing ESD have enabled en bloc resection of large lesions, even of whole circumferential lesions, and precise histological evaluation. Consequently, the indications for ER have gradually extended to include larger lesions that were previously treated surgically.

Many factors, including the patient’s condition, metastatic status, cancer invasion depth, tumor size, and tumor circumference, must be taken into account when choosing the appropriate treatment for SESCC. Among these factors, cancer invasion depth correlates well with the risk of metastasis and curability by ER [8,12], and tumor circumference is significantly associated with development of stricture following ER [13,14,15]. Therefore, accurate assessment of cancer invasion depth and tumor circumferential extent is crucial when selecting endoscopic treatment for patients with SESCC.

The indications for ER for SESCC of the Esophageal Cancer Practice Guidelines (2017) [16] are based on pathological findings. However, there can be considerable discrepancy between clinical and pathological diagnoses, especially concerning clinically (c)MM/SM1 and pathologically diagnosed (p)MM/SM1 cancers because the accuracy of endoscopic diagnosis of pMM/SM1 is relatively poor [17,18]. Considering that the indications for ER are determined on the basis of the clinical diagnosis, the applicability of ER should be investigated on that basis, not on the pathological diagnosis.

Furthermore, according to these guidelines, the extent of endoscopic resection is closely related to the risk of stricture. It is, therefore, strongly recommended that the circumferential extent of a lesion be evaluated preoperatively. It has also been reported that strictures are more likely to develop following ER when the lesion involves a large proportion of the circumference [13,14,15].

In this review, we evaluate the accuracy of preoperative diagnosis of depth of cancer invasion by various modalities by reviewing recent published reports and our institution’s data and discuss the applicability of ER according to the clinical diagnosis of SESCC.

## 2. Pretreatment Evaluations

Endoscopic examinations with white light imaging, narrow band imaging (NBI), and iodine staining are performed on all patients before treatments. The tumor depth is generally determined based on collaborative diagnosis of morphology with white light imaging and intrapapillary capillary loops (IPCL) with NBI magnification. Endoscopic ultrasound (EUS) of the esophagus and computed tomography scans are optional, and are usually performed only when endoscopy suggests invasion into the muscularis mucosae or deeper. However, it is not clarified whether the additional use of EUS allows more advantageous depth assessment to patients. Clinical implication to use EUS in addition to white light imaging and NBI magnification should be evaluated in the future.

Image-enhanced magnifying endoscopy or iodine staining is recommended to diagnose the lateral extent of the lesion, whereby the lesion border can be clearly delineated by the latter. However, use of iodine solution at a high concentration may cause the superficial epithelium to peel off, making a subsequent diagnosis difficult. Additionally, a prospective double-blind randomized controlled study reported that 1% iodine solution had a significantly lower pain score than 2% iodine solution, and the visibility of iodine-voiding lesions was the same in terms of color difference and physician assessment [19]. Hence, iodine solution is recommended to be used at a low concentration of ≤1% [20].

## 3. Accuracy of Preoperative Diagnosis of Depth of Cancer Invasion

Preoperative diagnoses of cancer invasion depth are used to divide lesions into three categories: tumor invades the epithelium or the lamina propria mucosae (EP/LPM), tumor invades the muscularis mucosae or the submucosa to a depth of 200 μm or less from the muscularis mucosae (MM/SM1), and tumor invades the submucosa to a depth of more than 200 μm (SM2).

The typical non-magnifying endoscopy (non-ME) appearances of each tumor category are as follows: cEP/LPM, flat lesion without protrusion or depression; cMM/SM1, flat lesion with irregular surface and protrusion <1 mm or shallow depression; and cSM2, lesion with protrusion ≥1 mm or deep depression. Endoscopic stage by magnifying endoscopy (ME)-NBI is classified according to the Japan Esophageal Society (JES) classification [18]. IPCL that show severe morphological changes are defined as type B vessels according to the JES classification, corresponding to squamous cell carcinoma [18]. Type B vessels are subclassified into three groups: B1, loop-like abnormal vessels showing dilation, tortuosity, caliber variation, and shape non-uniformity; B2, abnormal vessels without a loop-like formation; and B3, highly dilated, irregular vessels (irregular vessels more than about 60 μm in diameter, about three times thicker than B2 vessels). B1 vessels are defined as indicating EP/LPM invasion; B2 vessels, MM/SM1 invasion; and B3 vessels, SM2 invasion.

An area with no or low vascularity that is surrounded by type B vessels is defined as an avascular area (AVA). AVAs are classified into three groups according to size: AVA-small (<0.5 mm), AVA-middle (≥0.5 to <3.0 mm), and AVA-large (≥3.0 mm). These groups are defined as corresponding to EP/LPM invasion, MM/SM1 invasion, and SM2 invasion, respectively. However, an AVA surrounded by B1 vessels is defined as corresponding to EP/LPM invasion regardless of size.

The JES classification is used to make a comprehensive diagnosis on the basis of the evaluation of type B vessels and AVA.

Endoscopic stage by EUS is determined according to the destruction/preservation of the nine-layered structure as follows: cEP/LPM, lesion confined to the upper two layers with intact third and deeper layers; cMM/SM1, lesion that disrupts the first three layers with intact fourth and deeper layers; and cSM2, lesion with thinning or disruption of the fourth layer [21].

### 3.1. cEP/LPM Cancer

The positive predictive value (PPV) is reportedly 86–91% for cEP/LPM diagnosis based on non-ME [17,22] and 93–97% for diagnosis based on the JES classification [17,22], indicating very good results. Because the PPV of cEP/LPM diagnosis based on the JES classification is higher than that of diagnosis based on non-ME, the additional use of ME-NBI can be expected to enhance the accuracy of diagnosis of cEP/LPM. The sensitivity of Type B1 vessels for diagnosing pEP/LPM is also high [17,18,23,24]. Thus, identification of Type B1 vessels is useful for diagnosing pEP/LPM cancers.

The PPV for diagnosis of cEP/LPM cancers by EUS is 84%, which is lower than that for non-ME and ME [25]. Mizumoto et al. [25] reported that the sensitivity and accuracy with which ME-NBI distinguishes EP/LPM from MM/SM1 and SM2 are significantly higher than for EUS (83 vs. 72%, *p* = 0.048 and 82 vs. 70%, *p* = 0.017, respectively). Thus, EUS has limited additional value in patients with cEP/LPM identified by non-ME or ME.

### 3.2. cMM/SM1 Cancer

The PPV with which non-ME diagnoses cMM/SM1 is reportedly 53–65% [17,22] and 65–71% for JES classification diagnosis [17,22], which are relatively poor. The PPV for Type B2 vessels is insufficient, despite the high diagnostic yield for Type B1 and B3 vessels, because lesions with type B2 vessels ranges widely from pLPM to pSM2, with relatively high ratio in pSM2 cancers.

A previous study focused on the diameter of type B2 vessel area for misdiagnosis of SESCCs with type B2 vessels [26,27]. Takeuchi et al. [27] reported that the median diameter of type B2 vessel area in pSM2 was 10mm, which was significantly larger compared with that in pLPM cancers (5 mm) and pMM/SM1 cancers (4 mm). Adjusted by this factor, the PPV of type B2 vessels improved. Thus, lesions with a large type B2 area (≥10 mm) should be diagnosed as cSM2. On the other hand, Kimura et al. [26] showed that a type B2 vessel area diameter <6 mm and type B2 vessels around erosion were significantly associated with overdiagnosis in multivariate analysis. Therefore, lesions with a small type B2 area (<6 mm) and/or type B2 vessels around erosion should be diagnosed as cEP/LPM.

### 3.3. cSM2 Cancer

The PPV with which non-ME diagnoses cSM2 is reportedly 74–83% [17,22] and 77–93% for JES classification diagnosis [17,24], which are relatively good. Of note, the PPV of cSM2 diagnosis based on the JES classification is higher than that of diagnosis based on non-ME. Therefore, the additional use of ME-NBI is expected to improve the PPV for cSM2 cancer.

Although the PPV of cSM2 diagnosis based on the JES classification is excellent, the sensitivity is very low [16,18,27], being particularly low for Type B2 vessels with AVA-large (6–12%) [24,28]. To supplement the low sensitivity for Type B3 vessels and Type B2 vessels with AVA-large for SM2 cancers, Kimura et al. [26] reported Type B2 vessels with the findings such as nodular protrusion, thickness, and clearly depressed area, which are based on collaborative diagnosis of non-ME and ME-NBI, are reliable indicators of cSM2 invasion. Additionally, Matsuura et al. [29] have reported that Type B2 vessels on 0–I protrusions can be considered a criterion for cSM2 cancers.

## 4. Accuracy of Diagnosis of Cancer Invasion Depth According to Tumor Size or Circumference and Assessment of Curability

When determining the tumor depth based on collaborative diagnosis of non-ME and ME-NBI, we have to identify slight morphological irregularities and changes in IPCL or AVA, which can be accurately detected in small size lesions. However, it is more difficult to detect small infiltrating portions showing such irregularities and changes in entire area of larger lesions [28,30]. Thus, as the area of SESCC becomes larger, the rate of underdiagnosis is expected to increase, which is supported by our institution’s data analyzing diagnostic accuracy for cEP/LPM and cMM/SM1 cancers according to tumor size or circumference.

### 4.1. cEP/LPM Cancer

According to our institution’s data, the PPVs for comprehensive diagnosis (diagnosis based on non-ME and ME without EUS) for cEP/LPM in lesions <25 mm, ≥25 to <50 mm, and ≥50 mm are 94, 87, and 72%, respectively (*p* < 0.001) [30] (Table 1). Although the PPV decreases and the underdiagnosis of pMM/SM1 or pSM2 increases in parallel increasing lesions size, the PPV is acceptable at 70% even for cEP/LPM cancers ≥50 mm [31].

### 4.2. cMM/SM1 Cancer

According to our institution’s data, the PPV for the comprehensive diagnosis of cMM/SM1 in lesions <25 mm, ≥25 to <50 mm, and ≥50 mm in length is 44, 42, and 56%, respectively (Table 1). Comprehensive diagnoses were made mainly based on non-ME and ME, and EUS was conducted less than half of the cases.

When cMM/SM1 cancers are classified into three subgroups according to tumor circumference (≤3/4, >3/4 to <1, and whole circumference), the PPVs for comprehensive diagnosis of each subgroup are almost the same according to our institution’s data (Table 1 and Table 2). Remarkably, the underdiagnosis of pSM2 increases by 20, 33, and 36% in ≤3/4, >3/4 to <1, and whole circumferential extent, respectively.

## 5. Risk of Post-ER Stricture

Although ER is an effective treatment, extensive esophageal ER can lead to postoperative esophageal strictures. For example, in the absence of any preventive measures, the rates of postoperative stricture after non-circumferential resection and whole circumferential resection are 50–80% and 100%, respectively; and the required numbers of endoscopic balloon dilations (EBDs) are 6–9 and 22–33, respectively [32,33,34,35,36,37,38]. Stricture formation after esophageal ER causes dysphagia and requires multiple, long-term EBDs. It thus has a negative impact on patients’ quality of life and may delay additional CRT after non-curative resection [39]. Previous studies have shown that a mucosal defect >3/4 of the esophageal circumference after ER is a risk factor for stricture development [13,14,15].

Steroid therapy remains the mainstay of prevention of strictures. Local steroid injection is preferred for non-circumferential resection because of its efficacy (stricture rate: 4–45%), lower rate of complications, and convenience [33,36,40,41,42,43]. However, irrespective of preventive measures, whole circumferential resection is associated with an extremely high risk of stricture. It should, therefore, be primarily avoided whenever possible. When it is unavoidable, administration of oral steroids or a combination of oral steroids and local steroid injection may be the most effective strategies (stricture rate: 33–100% [35,38,44] and 18–92% [33,40,45], respectively).

## 6. Adequacy of Indications for ER for cEP/LPM, cMM/SM1, and cSM2 Cancers

As described above, the indications for ER are primarily determined on the basis of preoperative clinical assessment of cancer invasion depth and tumor circumferential extent, because cancer invasion depth correlates well with the risk of metastasis and curability by ER and tumor circumference is significantly associated with development of stricture following ER.

Previous guidelines for the diagnosis and treatment of SESCC recommend the following treatment strategies based on pathological findings: (1) ER is absolutely indicated for pEP/LPM cancers because of the extremely low risk of lymph-node metastasis, (2) ER is relatively indicated for pMM/SM1 cancers because the rate of lymph-node metastasis is 10 to 15%, and (3) SM2 cancers should be treated by surgery or definitive CRT because the rate of lymph-node metastasis is 30 to 50% [16,46,47,48].

On the other hand, as summarized in Table 2, our proposal for the indications for ER are based on clinical comprehensive diagnosis, not on pathological diagnosis, as follows. Previous studies [8,49,50] have reported that pEP/LPM cancers have a very low risk of metastasis and pMM/SM1 a low risk of metastasis, whereas pSM2 cancers have a high risk of metastasis.

### 6.1. cEP/LPM Cancer

Considering that more than approximately 90% of cEP/LPM cancers are finally diagnosed as pEP/LPM cancers, which have very low risk of metastasis, cEP/LPM cancers are a good indication for ER.

### 6.2. cMM/SM1 Cancer

The diagnostic accuracy for cMM/SM1 differs significantly with lateral spread of these lesions. In total, 80% of cMM/SM1 cancers with ≤3/4 circumferential spread are pEP/LPM or pMM/SM1 cancers, which have a very low or low risk of metastasis. Thus, cMM/SM1 cancers with ≤3/4 circumferential spread are adequate candidates for ER.

In contrast, ER is not recommended for whole circumferential cMM/SM1 cancers given the relatively low proportion of pEP/LPM or pMM/SM1 cancers and high risk of post-ER stricture. We recommend CRT or surgery to patients with whole circumferential cMM/SM1 cancers. As esophagectomy is associated with significant mortality and substantial morbidity, CRT is first recommended for patients who cannot tolerate surgery. ER may be an option for patients who would not receive CRT because of their wish to receive surgery as an additional treatment or poor condition. However, ER is likely to be non-curative resection, and delay of CRT due to refractory stricture must be taken into account when considering the use of ESD for whole circumferential cMM/SM1 cancers.

For cMM/SM1 cancers that involve >3/4 but not the whole circumference, ER should be considered on an individual basis because steroid therapy is effective in preventing stricture formation after non-circumferential resection, but the proportion of pEP/LPM or pMM/SM1 is relatively low at 67%.

### 6.3. cSM2 Cancer

Considering that 75% of cSM2 cancers are finally diagnosed as pSM2 cancers, which have a high risk of metastasis, cSM2 cancers are a contraindication to ER.

## 7. Conclusions

cEP/LPM cancers are a good indication for ER. ER is suitable as the first-line treatment for cMM/SM1 cancers with ≤3/4 circumferential spread, while ER is not recommended for whole circumferential cMM/SM1 cancers. For cMM/SM1 cancers that involve >3/4 but not the whole circumference, ER should be considered on an individual basis. cSM2 cancers are a contraindication to ER.

## 8. Future Perspectives

We have presented here the accuracy of diagnosis of SESCC invasion by assessing the findings of recent published reports and our institution’s data, and evaluated the indications for ER according to the clinical diagnosis of SESCC. However, all findings were retrospectively collected at single centers. Multicenter prospective studies are needed to confirm our findings and conclusions.

## Figures and Tables

**Table 1 jcm-10-00013-t001:** The positive predictive value (PPV) for comprehensive diagnosis of cancer invasion depth according to tumor size or circumference.

			pEP/LPM	pMM/SM1	pSM2
cEP/LPM ^†^	tumor size	~24 mm	94%	6%	0%
		25~49 mm	87%	11%	2%
		50~ mm	72%	22%	6%

cMM/SM1 ^††^	tumor size	~24 mm	40%	44%	16%
		25~49 mm	31%	42%	27%
		50~ mm	22%	56%	22%
	tumor circumference	≤3/4	36%	44%	20%
		>3/4 to <1	17%	50%	33%
		whole	14%	50%	36%
cSM2 ^†^			0%	25%	75%

^†^ Diagnosis based on non-magnifying endoscopy (non-ME) and magnifying endoscopy (ME) without endoscopic ultrasound (EUS)**.**
^††^ Diagnosis based on non-ME and ME, EUS less than half of the cases.

**Table 2 jcm-10-00013-t002:** The indications for endoscopic resection (ER) according to clinical diagnosis of cancer invasion depth and circumferential spread.

		pEP/LPM	pMM/SM1	pSM2	Risk of Stricture
cEP/LPM		90%	9%	1%	
cMM/SM1	≤3/4	36%	44%	20%	Low
	>3/4 to <1	17%	50%	33%	Moderate
	whole	14%	50%	36%	High
cSM2		0%	25%	75%	

Our proposal. 
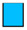
 Indication of ER with very low risk of metastasis. 
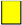
 Indication of ER with low risk of metastasis. 

 Contra-indication of ER with high risk of metastasis or stenosis.

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
