# Peer review of "Preoperative Diagnosis and Indications for Endoscopic Resection of Superficial Esophageal Squamous Cell Carcinoma"

_jcm, 2020, doi:10.3390/jcm10010013_

Round 1

Reviewer 1 Report

Page 2 line 68:

The use of EUS was optional. 

In general the use of EUS for esophageal is necessary for depth evaluation and to indicate those lesions that are amenable to endoscopic resection. The authors claim that EUS evaluation should be optional is not substantiated claim. 

Page 5 line 211:

Circumfrential lesions are not a contraindication to endoscopic submucosal dissection and resection. While they do carry a risk of stricturing, this can be endoscopically managed. There is no discussion in this article weighing the comorbidity of surgical resection. 

Author Response

Comments and Suggestions for Authors

Reviewer 1

Page 2 line 68:

The use of EUS was optional.

In general the use of EUS for esophageal is necessary for depth evaluation and to indicate those lesions that are amenable to endoscopic resection. The authors claim that EUS evaluation should be optional is not substantiated claim.

Response:

Thank you for important comments. The tumor depth is generally determined based on collaborative diagnosis of morphology with white light imaging (non-ME) and intrapapillary capillary loops (IPCL) with NBI magnification (ME-NBI). On the other hands, the PPV for EP/LPM cancers diagnosed based on EUS findings was 84%, which were lower than non-ME and ME-NBI (Mizumoto T, et al. Diseases of the esophagus 2018;31:7). Also, Mizumoto et al. reported that the PPV for MM/SM1 cancers was very low (29%), which was lower than that of ME-NBI. Goda et al (Goda K, et al. Diseases of the esophagus 2009;22:453). reported that the sensitivity and specificity of EUS in distinguishing mucosal (T1a-M) from submucosal (T1b-SM) cancers were 83% and 89%, respectively. Lee et al (Lee MW, et al. Scandinavian journal of gastroenterology 2014;49:853). showed that the accuracy of EUS in distinguishing M from SM cancers were 85%, which was higher than that of ME-NBI. Thus, additional use of EUS might be meaningful for estimating SM cancers, but effectiveness of EUS for each tumor category is not still clarified. So, we described “optional, not substantiated claim” and “are, not should be”.

Page 2, Lines 67-74: The first paragraph in the 2. Pretreatment evaluations section was revised as following: Endoscopic examinations with white light imaging, narrow band imaging (NBI), and iodine staining are performed on all patients before treatments. The tumor depth is generally determined based on collaborative diagnosis of morphology with white light imaging and intrapapillary capillary loops (IPCL) with NBI magnification. Endoscopic ultrasound (EUS) of the esophagus and computed tomography scans are optional, and are usually performed only when endoscopy suggests invasion into the muscularis mucosae or deeper. However, it is not clarified whether the additional use of EUS allows more advantageous depth assessment to patients. Clinical implication to use EUS in addition to white light imaging and NBI magnification should be evaluated in the future.

Page 5 line 211:

Circumfrential lesions are not a contraindication to endoscopic submucosal dissection and resection. While they do carry a risk of stricturing, this can be endoscopically managed. There is no discussion in this article weighing the comorbidity of surgical resection.

Response:

Thank you for valuable comments. As you pointed out, whole circumferential cMM/SM1 cancers are not a contraindication to ER. However, compared with non-circumferential resection, whole circumferential resection is associated with an extremely high risk of stricture irrespective of preventive measures. Refractory stricture may impair patients’ quality of life and may negatively affect the implementation of additional CRT, which is frequently applied for whole circumferential cMM/SM1 cancers composing of the relatively high proportion of pSM2 cancers. Thus, we consider that ER is not recommended for whole circumferential cMM/SM1 cancers. As described in introduction section, although esophagectomy is an effective treatment for esophageal cancer, it is associated with significant mortality and substantial morbidity. Thus, we first recommend CRT for patients who cannot tolerate surgical resection with whole circumferential cMM/SM1 cancers. ER may be one of the options for older patients or those with comorbidities who are not tolerant of even CRT. However, ER is highly likely to be non-curative resection and delayed additional treatment  due to refractory stricture must be taken into account when considering the use of ESD in whole circumferential cMM/SM1 cancers.

Page 5, Line 216-222: Following sentences were added in the 6.2. cMM/SM1 cancer subsection. “We recommend CRT or surgery to patients with whole circumferential cMM/SM1 cancers. As esophagectomy is associated with significant mortality and substantial morbidity, CRT is first recommended for patients who cannot tolerate surgery. ER may be one of the options for older patients or those with comorbidities who are not tolerant of even CRT. However, ER is likely to be non-curative resection and delay of CRT due to refractory stricture must be taken into account when considering the use of ESD for whole circumferential cMM/SM1 cancers.”

Reviewer 2 Report

This is a great insight into the complex classifications evolving in the world of early squamous cancers. It is important in that it tackles the real world problem of how to decide what should be endoscopically resectable based on the endoscopic findings alone. As the authors point out, there is no point calling SM2 pathology high risk when you don't know if it is SM" until you have taken it out.

Criticisms:

  1. Minor point is on lines 67-70:

"Endoscopic examinations with white light imaging, narrow band imaging (NBI), and iodine 67 staining were performed on all patients before treatments. Endoscopic ultrasound (EUS) of the 68 esophagus and computed tomography scans were optional, and were usually performed only when 69 endoscopy suggested invasion into the muscularis mucosae or deeper "

....use 'is' instead of 'were' because otherwise it sounds like you are embarking on a clinic study whereas this is a review.

2.The summary table at the end labels clinical definitions of the different stages but the clinical stage (eg cEP/LPM) is defined differently according to different modality (ie there is a cEP/LPM for NBI as well as for white light but these arent necessarily the same). How are the authors defining inthe cEWP/LPM in the final table (which is a table by the way and not a figure)

3. I'd like to see the lymph node rate per clinical stage in the table as it is easier to digest than from the writing).

Author Response

Comments and Suggestions for Authors

Reviewer 2

This is a great insight into the complex classifications evolving in the world of early squamous cancers. It is important in that it tackles the real world problem of how to decide what should be endoscopically resectable based on the endoscopic findings alone. As the authors point out, there is no point calling SM2 pathology high risk when you don't know if it is SM" until you have taken it out.

Response:

Thank you very much for your kind comments. We have responded to each suggestion and revised the manuscript accordingly.

Criticisms:

1.Minor point is on lines 67-70:

"Endoscopic examinations with white light imaging, narrow band imaging (NBI), and iodine 67 staining were performed on all patients before treatments. Endoscopic ultrasound (EUS) of the 68 esophagus and computed tomography scans were optional, and were usually performed only when 69 endoscopy suggested invasion into the muscularis mucosae or deeper "

....use 'is' instead of 'were' because otherwise it sounds like you are embarking on a clinic study whereas this is a review.

Response:

Thank you for pointing out the mistake. We corrected “were” to “are”. (Page 2, Lines 68 and 71)

2.The summary table at the end labels clinical definitions of the different stages but the clinical stage (eg cEP/LPM) is defined differently according to different modality (ie there is a cEP/LPM for NBI as well as for white light but these arent necessarily the same). How are the authors defining in the cEP/LPM in the final table (which is a table by the way and not a figure)

Response:

Thank you for important comments. The tumor depth is generally determined based on collaborative diagnosis of morphology with white light imaging and intrapapillary capillary loops (IPCL) with NBI magnification. In Table 2 (the summary table) as well as Table 1, we define cEP/LPM cancers based on the collaborative diagnosis.

Thank you for pointing out the mistake. We corrected “Figure 1” to “Table 2”. (Page 6, Line 230)

Page 2, Line 68-70: Following sentences were added in the 2. Pretreatment evaluations section. “The tumor depth is generally determined based on collaborative diagnosis of morphology with white light imaging and intrapapillary capillary loops (IPCL) with NBI magnification.”

Page 5, Lines 202-203: The third paragraph in the 6. Adequacy of indications for ER for cEP/LPM, cMM/SM1, and cSM2 cancers section was revised as following: On the other hands, as summarized in Table 2, our proposal for the indications for ER are based on clinical comprehensive diagnosis, not on pathological diagnosis, as follows.

3. I'd like to see the lymph node rate per clinical stage in the table as it is easier to digest than from the writing).

Response:

Thank you for your important suggestion. We show the rate of lymph node metastasis (LNM) per pathological depth of tumor invasion in our review (Page 5, Lines 196-201) based on previous reports dealing with only surgically resected cases with lymphadenectomy. However, we cannot show the rate of LNM per clinical diagnosis of depth because there are different treatments for each clinically diagnosed SESCC, such as endoscopic resection (ER), surgical resection and chemoradiotherapy (CRT). In cases treated with ER or CRT, we don’t know LNM at the point due to without lymphadenectomy. Thus, we are sorry that we cannot discuss about the rate of LNM in terms of clinical diagnosis.

This manuscript is a resubmission of an earlier submission. The following is a list of the peer review reports and author responses from that submission.